# Simulation-based reconstruction of global bird migration over the past 50,000 years

Marius Somveille[1,2,3,4]*, Martin Wikelski[3,5], Robert M. Beyer[6], Ana S.L. Rodrigues[7], Andrea Manica [6] & Walter Jetz [1,2]

Migration is a widespread response of birds to seasonally varying climates. As seasonality is particularly pronounced during interglacial periods, this raises the question of the significance of bird migration during past periods with different patterns of seasonality. Here, we apply a mechanistic model to climate reconstructions to simulate the past 50,000 years of bird migration worldwide, a period encompassing the transition between the last glacial period and the current interglacial. Our results indicate that bird migration was also a prevalent phenomenon during the last ice age, almost as much as today, suggesting that it has been continually important throughout the glacial cycles of recent Earth history. We find however regional variations, with increasing migratory activity in the Americas, which is not mirrored in the Old World. These results highlight the strong flexibility of the global bird migration system and offer a baseline in the context of on-going anthropogenic climate change.

[1] Ecology and Evolutionary Biology Department, Yale University, New Haven, CT, USA. [2] Center for Biodiversity and Global Change, Yale University, New Haven, CT, USA. [3] Max Planck Institute of Animal Behavior, Department of Migration, Radolfzell, Germany. [4] BirdLife International, The David Attenborough Building, Cambridge, UK. [5] Centre for the Advanced Study of Collective Behaviour, University of Konstanz, Konstanz, Germany. [6] Department of Zoology, University of Cambridge, Cambridge, UK. [7] Centre d'Ecologie Fonctionnelle et Evolutive CEFE UMR 5175, CNRS-Université de Montpellier-Université Paul-Valéry Montpellier—EPHE, Montpellier, France. *email: marius@somveille.com

Bird migration, which dramatically rearranges avian assemblages worldwide in direct response to seasonality[1–5], is a labile trait[6–9]. Previous research suggests that the original machinery of migration (physiological, behavioural, genetic)[10] evolved deep in the avian lineage[8,11] and its expression can change as a function of environmental conditions[12–17]. Accordingly, previous phylogenetic analyses found generally high (even if varying) rates of transition between sedentary and migratory behaviours and vice versa[18–20]. Analyses of current bird migration patterns have also shown that the seasonal ranges of migratory species, as well as the composition of avian communities in terms of migrant and resident species, are well explained by current climatic factors[5,21–24], suggesting that the global distribution of birds is approximately at equilibrium with current climate[23]. Consequently, on-going climate change is already affecting migration routes[14] and the prevalence of migrant species in avian assemblages[25]. Over time, climate change might, therefore, contribute to significant changes in global migration patterns, potentially leading to important net gains or losses of migratory behaviour in the avifauna.

Previous authors have thus hypothesised that the important variations in the Earth's climate during the glacial cycling of the Pleistocene had a major role in shaping current migratory pathways[7,26–29]. According to this hypothesis, the shifts and expansions of seasonal breeding ranges from glacial refugia into interglacial temperate regions, and away from non-breeding grounds, could have triggered a migratory behaviour in many species and shaped the global migration patterns observed today (e.g., by increasing migration distances). If this is the case, the importance of bird migration as an ecological phenomenon may be restricted to the warm interglacial periods (like the one we are in today), which are characterised by extensive areas featuring temperate climate conditions[30].

Understanding how the importance, prevalence and magnitude of migration across the avifauna vary throughout glacial cycles has relevance not just for understanding migration as a behavioural phenomenon, but also for gauging the past seasonal dynamics and the functional roles of birds in communities and ecosystems. However, addressing this problem directly is particularly difficult because the migratory behaviour is not recorded in the fossil record (i.e., fossils may indicate where species were present, but not if they migrated).

Previous species-specific studies have used analysis of genetic data and ecological niche modelling to investigate how the last glacial cycle has affected the evolution of migration, generally finding a strong effect[27–30]. However, recent simulation analyses of global bird migration[23,24] have suggested that, in addition to species-specific climatic niches, the seasonal redistribution of species is shaped by inter-specific competition for access to limited resources, for example associated with mutual interference[31,32], increasing search time[33] and territorial defence[34]. Investigating the past dynamics of bird migration can thus benefit from reconstructions at the scale of the avifauna, which go above and beyond independent, species-by-species models. A mechanistic model of the global seasonal distribution of birds—the Seasonally Explicit Distributions Simulator (SEDS)—has been recently developed to simulate spatial diversity patterns reflecting an equilibrium between the distribution of the global avifauna and climate[23]. This model relies upon the concept of energy efficiency (i.e., optimisation of energy budgets) to simulate the seasonal distributions of species, which can be sedentary or migratory. With the availability of climate reconstructions, this model provides a unique opportunity for simulating global bird migration in the past in order to investigate the response of the migratory avifauna to glacial cycles.

Here, we develop a new version of the SEDS model that integrates annual energy budgets explicitly (see details in Methods, and a complete description and discussion of the original model in ref. [23]). This framework is based on modelling the balance between costs and benefits of migration, with energy as a common currency, assuming a local carrying capacity to species richness based on primary productivity. The model simulates bird species' seasonal distributions—i.e., breeding and non-breeding ranges (coincident in resident species; different in migratory species)—that progressively saturate a virtual world with similar geography and seasonal distribution of energy supply as the real world. These virtual bird species distribute in a way that maximises energetic fitness, i.e., they maximise the amount of energy allocated to reproduction and survival by optimising the balance between energy assimilation and energetic costs associated with migration distance (both being a function of where the species' seasonal ranges are located), while taking into account the distribution of the other species, thus considering inter-specific competition for access to energy supply (Fig. 1, see Methods).

We apply the new version of the SEDS model to past climate data in order to simulate a reconstruction of the global seasonal distribution of birds over the past 50,000 years. This period encompasses the transition between the last glacial period and the current interglacial period, thus allowing us to investigate the effect of major climatic changes on the spatial patterns and importance of bird migration worldwide. Our results indicate that the prevalence of avian migration has remained largely stable across the globe over the past 50,000 years, albeit with noticeable geographical variations, which suggests that this phenomenon has been continually important throughout the glacial cycles of the Quaternary and that its origin might be more ancient.

## Results

**Predicting the current global seasonal distribution of birds**. The SEDS model simulates the distribution of the global avifauna in two seasons (capturing summer in the Northern Hemisphere, and summer the Southern Hemisphere) using simple rules reflecting a few key mechanisms that are derived from first principles of ecology and energetics, with only a single free parameter that could not be estimated directly from the literature (see Methods). Despite its simplicity, this model performs well in simulating the patterns associated with the current global seasonal distribution of birds, i.e., richness in breeding migrants, non-breeding migrants and residents (Supplementary Fig. 1). The model predicts particularly well the fact that breeding migrants concentrate around 50°N, with a strong asymmetry between the northern and southern hemisphere (correlation between empirical and simulated pattern = 0.795; Supplementary Fig. 1). It also correctly predicts the empirical observation that during their non-breeding season migratory birds largely concentrate in the southern part of the northern hemisphere (correlation between empirical and simulated pattern = 0.611; Supplementary Fig. 1). The pattern of resident bird diversity, with a peak in the tropics, is also well captured (correlation between empirical and simulated pattern = 0.637; Supplementary Fig. 1), even if the model underestimates the magnitude of this peak (leading to an underestimation of empirical total species richness). The model's good performance at simulating current breeding and non-breeding patterns of the global migratory avifauna supports an important role of energy efficiency (i.e., optimising the interplay between energy assimilation, which is affected by inter-specific competition for access to resources, and the energetic cost of travelling) in driving bird migration.

**Predicting the past global seasonal distribution of birds**. A mechanistic, simulation-based model with good explanatory

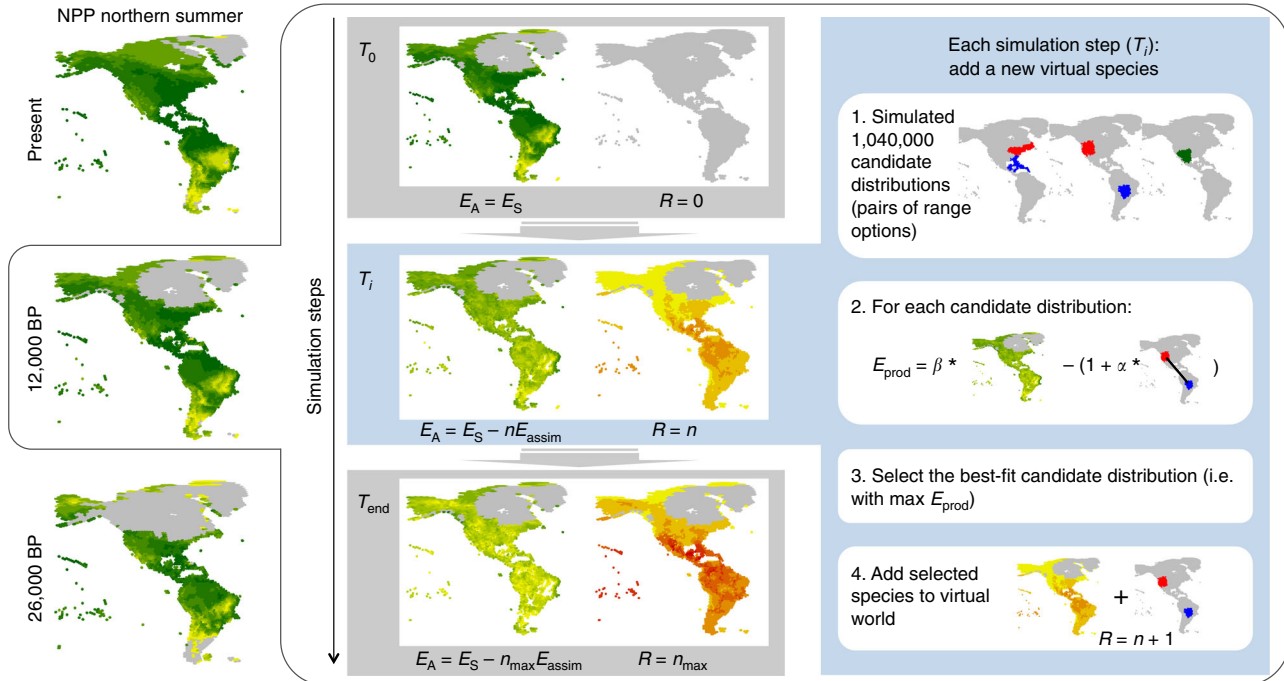

**Fig. 1 Model description.** The SEDS model was applied to each time slice of a climate reconstruction from present to 50,000 before present (BP); i.e., every 1000 years from present to 22,000 BP and every 2000 years before 22,000 BP. For a given time slice (example of 12,000 BP in this figure), the simulation starts ($T_0$) with a virtual world empty of bird species ($R = 0$). At this point, the energy available to birds is equal to the energy supply ($E_A = E_S$), estimated from the net primary productivity (NPP) obtained from the climate reconstruction. In each simulation step ($T_i$, sub-steps 1 to 4) a new virtual species is added to the virtual world. Its geographical distribution (combination of a breeding range and a non-breeding range) is selected among 1,040,000 candidate distributions, which are pairs of range options. Each range option is seeded randomly across the world and grown until a fixed size using a stochastic algorithm constrained by climatic conditions (see Methods; sub-step 1). The candidate distribution with the highest energetic fitness (i.e., maximum value of energy used for production, $E_{prod}$) is selected (sub-steps 2 and 3). $E_{prod}$ is computed as the energy assimilated by the species ($E_{assim}$), which is a linear function of the energy available within the geographical ranges ($\beta E_A$, type I functional response), minus energetic costs, which is equal to 1 (the basal energy used for survival) plus a linear function of the migration distance ($\alpha = 6.45e^{-5}$, see Methods). This way, the model optimises the balance between the energy assimilated through access to energy supplies and the energy used for travelling to determine the species' seasonal geographical distributions. As this new species is added to the virtual world ($R = n + 1$; $n$ indicating the total number of species at the start of a simulation step), the energy available $E_A$ is further depleted in the corresponding breeding and non-breeding ranges (sub-step 4). The simulation ends ($T_{end}$) when, after $n_{max}$ species are simulated, the maximum value for $E_{prod}$ is negative, meaning that any new virtual species would not have access to enough energy to compensate for the energetic costs associated with survival and therefore cannot exist.

power is particularly useful for making predictions into environmental conditions different from those in which the model was calibrated. Assuming that the apparent current equilibrium between climate and the distribution of the global avifauna equally applied to the past, we therefore used the SEDS modelling framework to simulate the global seasonal distribution of migratory birds through time (Fig. 1). We used a climate reconstruction covering the past 50,000 years (with 1000-year intervals between present and 22,000 years ago and 2000-year intervals earlier; Supplementary Movie 1) combined with a global vegetation model to obtain estimates of energy supply at regular intervals over that period (see details in Methods). When applied to environmental conditions over the past 50,000 years, our model predicts breeding distributions of migratory bird species progressively closer to the Equator, up to the Last Glacial Maximum (LGM, ~20,000 years ago), particularly noticeable in North America and the Western Palaearctic (Fig. 2). In particular, avian assemblages north of ~50°N are predicted to have been significantly poorer in breeding migrants than they are today, particularly prior to 10,000 BP (Fig. 2). We also predict that the geographical distribution of non-breeding migratory birds were concentrated closer to the Equator than at the present, although

this effect is less noticeable than for the breeding distributions (Fig. 2).

Our model predicts variations in the proportion of bird species that are migratory during the last glacial cycle. In the Americas, this proportion would have been ~20% smaller at the LGM than today (Fig. 3), corresponding to species that were resident during the last ice age and started migrating seasonally since then. A somewhat opposite trend is predicted to have occurred in the Old World, with the proportion of migrants similar to today or even higher during more ancient time periods (Fig. 3). A similar asymmetry is predicted in terms of migration distances. In the Americas, the model predicts that migration distance in the LGM was on average ~500 km shorter than today (Fig. 3), or conversely that today birds travel on average ~40% longer distances than they did at the end of the last ice age. In the Old World, however, the average distance travelled by migratory species is predicted to have slightly oscillated but remained on average largely stable over the last 50,000 years (Fig. 3).

Our model has no 'memory', in the sense that the global seasonal distribution of birds for any year is simulated independently of other time points. Despite this, it predicts a stable overall number of simulated bird species over the last 50,000 years (Supplementary Fig. 2) boosting confidence in the

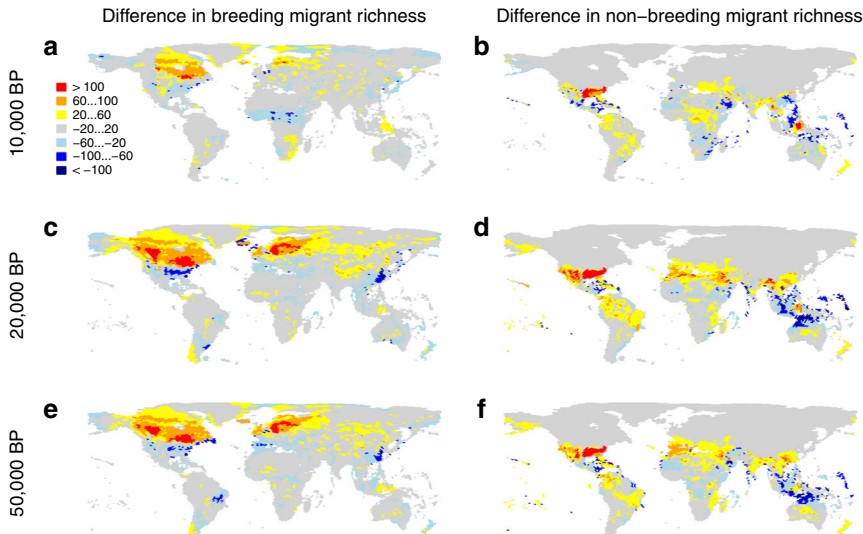

**Fig. 2 Contrast between past and present simulated patterns of migratory bird diversity.** The global patterns were computed as the predicted richness in the past, i.e., predicted number of species per hexagon: 10,000 years before present (**a, b**); 20,000 BP (**c, d**); and 50,000 BP (**e, f**), minus the predicted richness in the present. BP: before present. Red areas had more species than today, blue areas fewer.

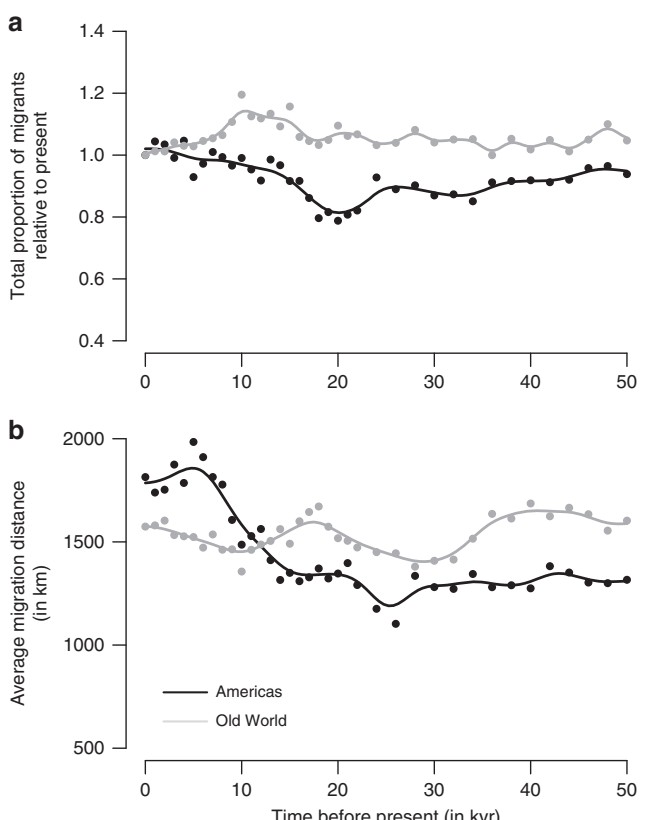

**Fig. 3 Predicted proportion of migrants and average migration distance over the last 50,000 years. a** Evolution of the total proportion of simulated bird species that are migrants across the world relative to the present value. **b** Evolution of the average distance between breeding and non-breeding grounds for migrant species, computed as the great circle distance between the centroids of the seasonal ranges. These simulated time series are shown for the Americas (in black) and the Old World (in grey). 1 kyr = 1000 years.

model predictions, because it is indeed not expected that the number of avian species changed much over that period.

## Discussion

Overall, our findings suggest that throughout the last 50,000 years, spanning the last glacial maximum (~20,000 years ago), bird migration remained an important global phenomenon, refuting the hypothesis that this is mainly a phenomenon of interglacial periods during which the planet features large areas with temperate climate conditions[29,30]. This contrasts with previous results by Zink and Gardner[30], who predicted from species-specific climate niche models for 56 North American migratory species that most species were sedentary during the last glacial maximum and that glaciations are major 'migratory switches'. Our results, which are based on simulations at the avifauna-scale, point instead to an origin of migration over a much longer time scale than the glacial cycles of recent Earth history[18–20,35–37].

Our simulations also indicate that the magnitude of the avian response in terms of migratory behaviour to past global change was likely variable across the world. North America is the region of the world that is predicted to have seen the greatest changes in bird migration since the last ice age, alongside the retreat of the large Laurentide ice sheet. In this region, we predict a southwards compression of bird migration as we go back in time (Fig. 4), particularly of breeding ranges (Fig. 2), with a predicted shift of the transition zone between southern avian assemblages that are net senders of breeding migrants and northern assemblages that are net receivers of breeding migrants from ~35°N today to <30°N at the LGM (Fig. 4). This corresponds to a significant increase in the average migration distances covered in this region since the LGM (~40% increase) and in the proportion of species that are migratory (~25% increase; Fig. 3). In the Old World, the western Palaearctic is also predicted to have experienced relatively important changes alongside the retreat of the ice, with seasonal grounds of migrants at the LGM being closer to the equator than today (Fig. 2). This is somewhat less pronounced than in North America, which is in line with the relatively smaller extent of the Eurasian ice sheets, and we predict that it had little effect on the

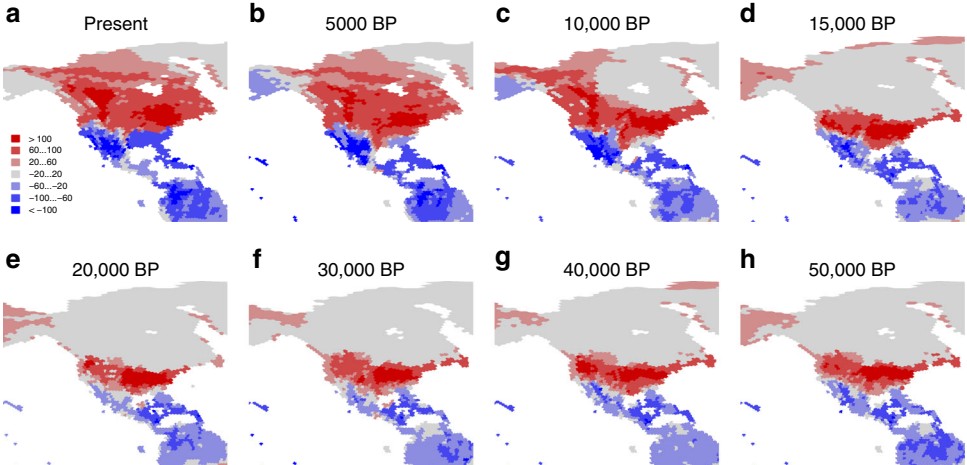

**Fig. 4 Fifty-thousand years of predicted seasonal difference in richness due to migration, across the northern Western Hemisphere.** The maps **a–h** show the seasonal difference in avian richness, computed as the richness in breeding migrants minus the richness in non-breeding migrants, for eight time slices between the present and 50,000 years ago. BP: before present. Red colours are regions with avian higher richness during the breeding season; blue colours are regions with higher richness during the non-breeding season.

proportion of species that are migratory or in the average distance they travel (the former even appearing to be somewhat higher in the past than today; Fig. 3 and Supplementary Fig. 2).

In our simulations, patterns in the seasonal redistribution of the world's avifauna emerge from the optimisation of energy budgets as birds use migration as a strategy to maximise energetic fitness. The number of current resident species, however, is substantially underestimated, particularly around the equator (Supplementary Fig. 1), which results in a too high fraction of migrants in relation to resident species across the world (Fig. 3 and Supplementary Fig. 2). A significant amount of the variation in the seasonal distribution of the migratory avifauna also remains unexplained by the model (Supplementary Fig. 1). This suggests that additional mechanisms need to be included in future versions of the model in order to better explain the empirical diversity patterns. This may include more realistic approaches to modelling migratory costs than the shortest distance between seasonal grounds, by taking into account geographical features (e.g., seas, mountains, distribution of stopovers[3,38–40], wind patterns[41], and predation risk[42]. Moreover, the model currently treats all species as equivalent and equally abundant locally, but differences in species' evolutionary history and ecology could also be important for explaining global empirical patterns. For example, we underestimate the number of species across high-latitude northern temperate regions over winter, yet several bird lineages (e.g., parids, corvids, woodpeckers, finches) have evolved adaptations other than migration to increase survival in these seasonally energy-depleted environments[43]. Furthermore, the current version of the model does not perform as well at predicting the proportion of all species that are migrants as the original SEDS model[23], suggesting that the model of net primary productivity we applied here (one that can be projected back in the past, see Methods), might not be as adequate at estimating energy supply as the remote-sensing data used in the original SEDS model.

Given that shifts in the spatial distribution of biodiversity are the combined result of individual species' responses[44], a further development would be to apply our global model in tandem with single species models, the former being used to model the background community and the latter for making species-specific predictions on the evolution of their distributions and migration since the last glacial period. This would allow investigating for example if rapid transitions between being sedentary and being migratory and vice versa occurred since the last ice age, something that cannot be captured by our global model alone. Estimating the rate of species-specific gains and losses of migratory behaviour due to glacial cycling could in turn inform phylogenetic analyses over evolutionary time (e.g., refs. [18–20,37]) and thus bring new insights into the origin and evolution of migration.

Our results suggest that bird migration systems across continents have not responded the same to past climate change. The differences in the past waxing and waning of the migratory phenomenon between continents can potentially explain patterns observed today, such as differences in migration strategies between avifaunas. For example, the need for communication calls during migratory flights might be higher in the New World[45] to compensate for the fact that species' migratory behaviours have been particularly variable over time.

The rapid anthropogenic climate change that Earth is currently experiencing is likely to have a strong impact on the distribution and movement of species and biodiversity. While non-mobile species will likely have to locally adapt to change, highly mobile species might be able to move and track changing environmental conditions. In this context, the magnitude and flexibility of the response of bird migration to global change highlighted by our results offers a baseline for predicting how migratory birds will respond to future climate change.

## Methods
**Climate reconstruction.** Monthly climate data (temperature, precipitation and cloud cover) covering the past 50,000 years were obtained from the HadCM3 global circulation model[46]. These data are at 1000-year intervals between present and 22,000 years before present (BP) and at 2000-year intervals between 22,000 and 50,000 BP. The original simulation data on a 2.5° × 3.5° were bias corrected and downscaled using the delta method[47], which builds a difference map between simulated and observed data (in our case, high-resolution present-day estimates from ref. [48]) and uses it across time intervals. This approach has been shown to be the most robust solution to debias paleoclimatic reconstructions for the late Pleistocene[49]. We first downscaled our paleoclimatic variables to a 1/6 degree grid, and subsequently remap them onto a global grid of equal-area, equal-shape hexagons (internode spacing of ~153 km). With the HadCM3 simulations, we used the global ice sheets reconstruction data set ICE-6G version 1.2[50]. Hexagons covered by ice sheets were considered not habitable for all birds, regardless of climatic conditions. Seasonal climate values for temperature and precipitation were obtained by averaging the monthly climate values over the northern winter (November to February included) and the northern summer (May to August included). Visualisations of the climate reconstructions are presented in Supplementary Movie 1.

**Model overview**. We developed a new version of the Seasonally Explicit Distributions Simulator (SEDS) model, which was originally described and discussed in ref. [23]. Here, we integrate energy budgets more explicitly and we reduce the number of free parameters. The SEDS model is based on modelling the balance between costs and benefits, with energy as a common currency. It is built on three main components: (1) a set of virtual species' range options, (2) the estimated energy requirements associated with key biological processes, and (3) the estimated spatial and seasonal variation of the energy supply available to birds in the environment. Integrating these three components, the model is applied through a sequence of simulation steps whereby a virtual world—with the same geography and seasonality as the real world, mapped onto the global hexagonal grid described above—is filled with virtual species until it becomes saturated. In this model, virtual bird species are functionally equivalent (i.e., we ignore differences in traits and ecology) and are represented by a combination of a breeding range and a non-breeding range that can be either congruent (resident species) or different (migratory species).

At the start of a simulation, the virtual world is empty of bird species, and each simulation step consists of adding a virtual species into it. The geographical distribution of each virtual species (i.e., combination of breeding and non-breeding ranges) is selected among candidate distributions as the one with the maximal energetic fitness (Fig. 1). The newly simulated virtual species then consumes energy within its geographical distribution equivalent to its corresponding energetic cost, effectively depleting the energy available in all the hexagons across its geographical distribution. We stop simulating species when the virtual world is saturated with simulated species. Each simulation was performed separately on the Western Hemisphere (WH < 30°W) and Eastern Hemisphere (EH > 30°W).

The model is mechanistic in the sense given by Connolly et al.[51]: 'a characterisation of the state of a system [here, the global seasonal distribution of birds] as explicit functions of component parts [species' geographical ranges optimising energy budgets] and their associated actions and interactions [inter-specific competition for access to energy supply]'.

**Virtual species range options**. For each time slice separately, we generated 1000 contiguous geographical ranges in our virtual world (400 in the WH, 600 in the EH, reflecting differences in area) to serve as options from which the distributions of virtual species were simulated (Fig. 1). These range options all had a size of 131 hexagons in the western hemisphere and 180 hexagons in the eastern hemisphere, which correspond to respective median values in a global data set of avian species' range maps[52]. Ranges were generated using a method adapted from the spreading dye algorithm[53,54] through a climate-driven approach of range expansion that has been shown to accurately capture the empirical distribution of bird ranges' shape[55]. Each range was seeded from a single hexagon, randomly selected among all hexagons each with a probability $P_h = 1/(1 + S_h)$ (Eq. (1)), with $S_h$ denoting the number of species already simulated and occurring in hexagon $h$. The probability of selecting a given hexagon as a seed hexagon was thus a function of the local richness in virtual species in order to avoid simulating range options that are too clustered spatially. From the selected seed hexagon, we then allowed a stochastic spread into adjacent unoccupied hexagons, constrained by climatic conditions, until the virtual range reached a fixed size. For each range, an initial climatic optimum was obtained from the position of the seed hexagon in a climatic space defined by a mean annual temperature (z-standardised) and a mean annual precipitation (z-standardised after being log-transformed). We then selected two neighbours of the seed hexagon, with the probability of selection being higher for neighbours closer to the climatic optimum (that is, lower Euclidian distance $d$ in the climatic space between itself and the climate optimum), calculated as $P_{select} = 2(d + 1)^{-30}$ (Eq. (2)), divided by the sum of these values across all neighbours (hence decaying exponentially with increasing climatic distance $d$). We then repeated this procedure, each time redefining the climatic optimum as the average climatic condition across the already selected (that is, occupied) hexagons and selecting 25% (rounded to the larger integer) of the set of unoccupied neighbours of the occupied hexagons (summing the probabilities of the ones being neighbours of more than one occupied hexagon), until the desired range size was reached (131 hexagons, ~3.05 million km², in the WH, 180 hexagons, ~4.20 million km², in the EH). Visualisation of the geographical distribution of range options through time is presented in Supplementary Movie 2.

**Energy supply**. In each hexagon, the energy supply $E_S$ is the total amount of available resources that can be used to fuel bird species' metabolism. In ref. [23], it was modelled as proportional to the Normalised Difference Vegetation Index (NDVI), but NDVI is a remote-sensing measure of greenness of the land that cannot directly be reconstructed in the past. Therefore, here, we modelled energy supply as proportional to net primary productivity (NPP). Monthly NPP for each time slice was estimated with the Biome4 global vegetation model[56] using monthly temperature, precipitation and cloud cover reconstructions (see above) as inputs (Supplementary Movie 1).

We assumed that in any given hexagon $j$, the carrying capacity for birds is proportional to NPP, such that $E_{S_j} = \mu * log_{10}(NPP_j + 1)$ (Eq. (3)). Negative NPP values were set to zero. The parameter $\mu$ was used for adjusting the energy supply

(that is, acting as carrying capacity) for the model to generate a realistic total number of virtual species. In this study, however, we were not interested in replicating precisely the total number of bird species occurring in the world, but rather in investigating how the spatial patterns associated with the global seasonal distribution of birds changed in the past. We thus used a fixed value of $\mu = 65$ for all of our simulations. This value for $\mu$ was chosen, after looking at the range of values for NPP, to obtain simulated species richness that are in the same order of magnitude as the empirical data. We conducted a sensitivity analysis to make sure that this value for $\mu$ was not crucial for the results (see section on sensitivity analysis below). The energy available in any given hexagon $E_{A_j}$ in a given simulation step is equal to the energy supply minus the energy already consumed by species simulated during previous simulation steps and occurring in hexagon $j$. The energy available to a species at a given season (breeding or non-breeding) was computed as the mean of the energy available in all the hexagons across the seasonal range.

**Energetic costs**. The energetic requirements associated with a species' seasonal range were modelled as a function of two terms: the basal energy use for existence, $BE_U$, which was set to be 1 (arbitrary) unit of energy use, and the additional cost associated with migration ($m_C$), which was converted into these same (arbitrary) units. The cost of migration corresponds to the energetic cost of, each year, travelling between the breeding and non-breeding ranges. We assumed that $m_C$ increases linearly with distance travelled (thus, $m_C = 0$ for resident species), and migration happens instantaneously at the end of each season (its cost added to the corresponding season to reflect the previous investment in fat reserves). For each season, $m_C$ was computed as a function of the great circle distance, $d_m$, between the centroids of the breeding and non-breeding geographical ranges (average distance travelled by individual birds of the species assuming that they migrate using the shortest route). Thus, $m_C = \alpha * d_m$ (Eq. (4)), with $\alpha$ being a parameter determining the energy required for a bird to travel a unit of distance.

The parameter $\alpha$ can be estimated directly from the literature on flight physiology[57] as equals to flight power ($F_W$, in J/s) divided by flight speed ($F_S$, in m/s). To rescale the cost of migration in terms of the arbitrary units of energy use, we compared the energy used for the migratory journey to the basal metabolic rate (which approximates minimum levels of energy expenditure for existence) over a whole season ($BMR_S$, in J), such that: $\alpha = \frac{F_W}{F_S BMR_S}$ (Eq. (5)). Detailed comparative studies found that $F_W$ and $F_S$ scale with body mass ($M$, in g) such that $F_W = 0.257M^{0.763}$ (Eq. (6)) (estimated using data from ref. [58] on the cost of forward flapping flight for 31 avian species, excluding seabirds) and $F_S = 6.4773M^{0.13}$ (Eq. (7)) (estimated by ref. [59] measuring the cruising speed of 138 species of migratory birds in flapping flight), respectively. We used the allometric relationship for the basal metabolic rate (BMR, in mlO₂/h) described for 211 avian species in ref. [60] as: $BMR = 6.7141M^{0.6452}$ (Eq. (8)), which we then converted to J/s using the conversion factor $1J/s = 172mlO_2/h$, and multiplied by the number of seconds in 6 months (i.e., ~15,724,800) to obtain $BMR_S = 6.15e^5M^{0.6452}$ (Eq. (9)). The resulting estimation for $\alpha$ was therefore approximately independent of body mass, with the cost of migration equal to: $m_c = 6.45e^{-5}d_m$ (Eq. (10)), with $d_m$ the travel distance in kilometres. This corresponds to an energetic cost for migration of ~0.065 or ~6.5% of the yearly basal energy use for existence if the species travels an average of 1000 km between its breeding and non-breeding geographical ranges.

**Maximising energetic fitness**. As a model development in relation to ref. [23], here we modelled fitness explicitly, using an energetic definition (e.g., ref. [61]). Birds assimilate biochemical energy initially converted mostly from solar radiation energy via photosynthesis. This assimilated energy ($E_{assim}$) fuels two main metabolic processes: respiration, which powers the work of living, and production, which generates new biomass. Using energy as the common currency, it translates into two components of fitness: energy used for survival ($E_{surv}$) and energy used for production ($E_{prod}$). The following relationship can be derived from this definition:

$$E_{prod} = E_{assim} - E_{surv} \qquad (11)$$

We assume that, to maximise fitness, birds maximise $E_{prod}$ on an annual basis. For each virtual species to be simulated, we therefore looked for the candidate distribution (i.e., combination of virtual seasonal range options) with the highest associated value for year-round $E_{prod}$. To do so, for each candidate distribution, we estimated annual $E_{assim}$ and $E_{surv}$. To estimate $E_{assim}$ during a given season, we assumed a type I functional response of birds to the energy supply available in the environment ($E_A$) as:

$$E_{assim} = \beta E_A \qquad (12)$$

where $\beta$ is a parameter governing this linear relationship.

To estimate $E_{surv}$ during a given season, we computed the sum of the basal energy use for existence ($BE_U$), which was set to 1 arbitrary unit, and the energetic cost of migrating between the seasonal distributions (see details in the section on energetic costs above), as:

$$E_{surv} = BE_U + \alpha d_m = 1 + 6.45e^{-5}d_m \qquad (13)$$

The year-round amount of energy allocated to production for a given candidate distribution was therefore obtained using the following formula:

$$E_{prod} = \beta\left(E_a^{(NS)} + E_a^{(NW)}\right) - 2(BE_U + \alpha d_m) \tag{14}$$

where NS indicates northern summer, and NW indicates northern winter.

The geographical distribution (i.e., combination of breeding and non-breeding virtual ranges) of the virtual species to be simulated was selected as the candidate distribution with the highest $E_{prod}$ (i.e., the highest energetic fitness). This candidate distribution was thus the one with an optimal balance between the energy assimilated through access to energy supplies and the energy used for travelling. The breeding season was set to be the season with the highest $E_{prod}$, essentially assuming that species maximise the amount of energy directly allocated to reproduction, and the geographical range associated with this season was therefore assigned as the breeding distribution of the species while the other range was assigned as the non-breeding distribution of the species. The newly simulated virtual species consumes energy within its seasonal geographical ranges equivalent to the corresponding $E_{assim}$, effectively depleting the energy available in all the hexagons across its geographical distribution. We stopped simulating species when the $E_{assim}$ value for the selected distribution was below $BE_U$, meaning that the species could not assimilate enough energy to fuel the basal energy use for survival. This indicates that the virtual world is saturated with simulated species so that no new species can be added to it and survive.

**Global patterns in the seasonal distribution of birds**. The SEDS model outputs virtual species seasonal distributions across the world, from which global diversity patterns can be mapped. We generated the following three basic spatial patterns that captured the global seasonal distribution of terrestrial birds: 'richness in breeding migrants', the number of species present in each hexagon only during their breeding season; 'richness in non-breeding migrants', the number of species present in each hexagon only during their non-breeding season; and 'richness in residents', the number of bird species present in each hexagon year-round. In parallel, we have also quantified these patterns using empirical data on bird distribution: spatial polygons representing the global distribution of 9783 non-marine bird species, obtained from BirdLife International and NatureServe[52]. The data and their treatment for generating these global richness patterns are described in detail in ref. [23].

**Parameters scan**. The improved SEDS model used in this study has only one free parameter that could not be estimated directly from the literature: $\beta$, which determines the type I functional response between energy available in the environment and energy assimilation. We explored the following range of values for this parameter:

$\beta \in \{0.003, \ldots, 0.035\}$ with a step of 0.001.

Values below 0.003 resulted in not even one virtual species being able to survive as the energy it assimilated was already below its energy requirement for survival. Also we bounded the range of values to an upper limit of 0.035 because above this value the energy assimilation for the first simulated species (i.e., with maximum possible energy available) became highly unrealistic: >17 times the basal energy used for survival.

To assess the quality of the model outputs for each simulation given a $\beta$ parameter value, we computed the correlation coefficient between empirical and simulated patterns for the global seasonal distribution of birds by summing correlation coefficients for the three patterns described above: richness in breeding migrants, richness in non-breeding migrants and richness in residents.

Low values of $\beta$ lead to poor model performance, i.e., low correlation between the simulated global patterns and the empirical ones (Supplementary Fig. 1), as well as a very low number of species generated. For $\beta > 0.01$ the performance of the model plateaus above a mean correlation of 0.6 between empirical and simulated patterns, indicating fairly good performance of the model (Supplementary Fig. 3). However, for $\beta > 0.015$ the correlation tends to slightly decreases as $\beta$ increases. The total species richness generated also peaks between $0.01 > \beta > 0.015$, even though it does not go above 4000. This is less than half the actual number of bird species in the world. For every value of $\beta$ investigated, the model also predicts a total proportion of migrants in the global avifauna that is much higher than the real one ( > 45% vs. 15%, respectively). We selected $\beta = 0.012$ as our best-fit (i.e., most realistic) value to be used for back-casting the global seasonal distribution of birds. This value gives the best compromise between maximising the match to empirical patterns, generating the maximum number of species and minimising the total proportion of migrants (Supplementary Fig. 3).

**Sensitivity analysis**. We explored the robustness of the results obtained for the best-fit model by running the model with varying values for $\alpha$, the parameter associated with the cost of migration previously estimated directly from the literature, with $\alpha \in \{0.00001, \ldots, 0.0005\}$, and $\mu$, the parameter for rescaling NPP into energy supply, with $\mu \in \{45, \ldots, 150\}$. Each time, we ran the model keeping all other parameters fixed and investigated the model performance at re-producing the empirical patterns associated with the global seasonal distribution of birds.

We also explored the sensitivity of the model to the way we simulated geographical range options. We investigated the model performance using different values of range size (i.e., number of hexagons $\in \{100, \ldots, 200\}$ for range options in the western hemisphere, and number of hexagons $\in \{150, \ldots, 250\}$ for range options in the eastern hemisphere), as well as varying the strength of the climatic constraints when growing range options, i.e., varying the probability of selecting neighbours calculated as $2(d+1)^{-x}$ by investigating values for $x \in \{1, \ldots, 45\}$.

The model performances were very robust to variations in $\mu$, the parameter for rescaling NPP into energy supply for birds (Supplementary Fig. 4), as well as to how range options were simulated (Supplementary Figs. 5 and 6). Varying $\mu$ did not affect much the ability of the model to predict the current global seasonal distribution of birds (i.e., correlations between simulated and empirical patterns always >0.6), and did not affect the total proportion of migrants simulated (Supplementary Fig. 4). However, when $\mu$ increases then the total number of species generated increases almost linearly, which is not surprising since $\mu$ determines the total amount of energy supply, and thus the carrying capacity of the virtual world. Varying the size of the geographical range options simulated, as well as the strength of the climatic constraints determining their shape, did not affect the model performance (i.e., correlations between simulated and empirical patterns always >0.6) nor the total proportion of migrants simulated (Supplementary Figs. 5 and 6). Increasing the size of the simulated range options led to a nearly linear decrease in the total number of simulated species as less species can be fit in the virtual world for a given total carrying capacity. In contrast to the other sensitivity analyses, variation in $\alpha$, the parameter determining the cost of migration, did affect significantly model performances, with relatively high values ($\alpha > 0.001$), leading to the model poorly capturing the current global seasonal distribution of birds and model performance also decreasing with decreasing $\alpha$ below $\alpha = 0.0005$ (Supplementary Fig. 7). The direct estimation of $\alpha$ from the literature ($\alpha = 0.000645$, see section on energetic costs above) is within the peak of model performance (i.e., $0.0004 > \alpha > 0.0008$), which boosts the realism of the model. This peak of model performance also corresponds to a dip in the total number of simulated species, although the latter does not vary much with varying $\alpha$ (Supplementary Fig. 7). The total proportion of migrants decreases almost linearly with increasing $\alpha$, which is not surprising as this corresponds to an increase in the cost of migration.

We also reconstructed the global seasonal distribution of birds over the last 50,000 years for several combinations of parameter values other than the best-fit model. In addition to three alternative values for $\beta$, we investigated the outputs for three alternative values for $\alpha$ and three alternative values for $\mu$, selected as performing relatively well for the present. The predicted evolution of global migration over the last 50,000 years is fairly robust to variation in values for parameters $\beta$, $\alpha$ and $\mu$. The total number of species simulated remains stable over the last 50,000 years, with a slight decrease between 10,000 BP and 20,000 BP, for every combinations of parameter values investigated (Supplementary Figs. 8–16). The total proportion of migrants among simulated species also remains relatively stable over the last 50,000 years, with a slight decrease often observed in the Americas between 10,000 BP and 20,000 BP, for every combinations of parameter values investigated (Supplementary Figs. 8–16). The model exhibits a very similar pattern through time for the average migration distance among simulated migratory species for every combinations of parameter values investigated, being very stable in the Old World while decreasing by ~200–700 km between the present and the LGM in the Americas (Supplementary Figs. 8–16), except for when the cost of migration is very high ($\alpha = 0.005$; Supplementary Fig. 13) as migratory species already travel very short distances to avoid the high energetic costs.

## Data availability

The bird species distribution data are available for non-commercial use upon request to BirdLife International (http://datazone.birdlife.org/species/requestdis). Monthly climate and vegetation data covering the past 50,000 years are available on Open Science Framework at DOI 10.17605/OSF.IO/9CSJA. The ice sheets data ICE-6G version 1.2 are freely available to download at https://atmosp.physics.utoronto.ca/~peltier/data.php.

## Code availability

The computer code used for this study is available from the corresponding author upon request.

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

## Acknowledgements

We are grateful to BirdLife International, NatureServe and all the volunteers who collected and compiled the data on the empirical distribution of bird species. This research was supported through the Max Planck—Yale Centre for Biodiversity Movement and Global Change and from the Knobloch Family Foundation, as well as grants NSF DEB 1441737, DBI 1262600 and NASA NNX11AP72G to W.J., A.M. and R.M.B. were supported by the ERC Consolidator Grant 647787 ('LocalAdaptation').

## Author contributions

M.S. conceived and developed the model, performed the analyses with support from A.M. and W.J., and drafted the manuscript. R.M.B. and A.M. contributed the climate reconstructions. M.W., A.S.L.R. and all the other authors, provided critical input on the manuscript.

## Competing interests

The authors declare no competing interests.
