## [Peer Review File · Nature Communications]

Reviewers' comments:

Reviewer #1 (Remarks to the Author):

This is an interesting and well-written paper that tests whether or not birds were likely to have been migratory throughout the last 50,000 years of earth history, given shifting climatic conditions and glacial maxima. The research builds on a previously published model used to simulate the seasonal geographic distributions of birds in relation to global energy supply, and uses present-day model predictions to infer past conditions. The extent to which birds migrated during the last glacial maximum has been the subject of some recent debate, and this paper is by far the most sophisticated analysis of the question to date.

General comments:

1) I liked the paper and generally found it readable and accessible, with a few exceptions that I note below. My main comment is that the entire modeling framework is quite complex as a whole, and relies on a large number of moving parts and building blocks. Some of these parts are well-described in this paper, and for others the reader has to look back to the previous paper describing the original model (which I did). Some of the methods used are intuitive, and others are very specialized methods with which a typical reader is unlikely to be familiar (e.g., 'spreading-dye algorithm, 'earth's-mover's distance,' the 'delta-method.'). I applaud the author's creativity and ingenuity to find methods that they could apply to each component of their model, and also their efforts to look to the biological literature to come up with best-guess parameter estimates. They have made a strong effort to build a model to test their hypotheses in a reasonable and sophisticated way. Nevertheless, putting everything together amounts to a black box in which it is essentially impossible for a reader to understand how particular modeling or methodological decisions affects their results. Their sensitivity analyses help with this to some extent, but not entirely. This complexity is not necessarily a bad thing, but it demands some care when describing and interpreting the study. What I think is mainly needed now is more contextualization of their study as a simulation, because essentially that's what they're doing -- simulating a virtual world. So, for example, Line 8 in the abstract should read, "we apply a mechanistic modelling framework to climatic reconstruction data to SIMULATE the effect of major climatic changes on the spatial patterns..." To me, this gives a better description to the reader of what is really happening. Similarly, throughout the paper the authors should be more careful not to take the results of the original model (reference 22) as established fact, because, again, that study is a simulation and in fact the current model is pitched as an improvement over the previous one. I should be clear that what I'm suggesting are relatively minor changes to descriptive language, but I think they are important.

2) I think the authors should consider that migration is just one of several adaptations that allow for organismal persistence in a seasonal environment and think about how this influences their interpretation (see, for example, Winger et al. 2019 Biological Reviews (doi: 10.1111/brv.12476), for a detailed discussion of this). The reason this is important to discuss is that alternative adaptations

to seasonality (e.g., hibernation of mammals, deciduous trees dropping leaves) lead to a different distribution of biodiversity throughout the year, and yet all of these organisms are dealing with the same seasonal flux and spatial distribution of global energy. Below, I point out a couple of specific parts of the text that would be improved by considering this interpretation of migration as one type of persistence strategy:

Line 133: "In our reconstruction of global bird distributions, migration appears as an emergent phenomenon resulting from the maximisation of species' energetic fitness, with spatial diversity patterns reflecting an equilibrium between the distribution of the global avifauna and climate." I think this characterization needs further thought. The results, in my opinion, do not show migration itself as an emergent phenomenon of energetic fitness per se. Rather, the seasonal distributional patterns of birds are the emergent response to energy, given that birds use migration as their principal strategy for maximizing energetic fitness. I.e., if more lineages of birds had evolved alternative strategies for coping with low energy environments, their distributional patterns would be different. The depauperate winter avifaunas of the northern hemisphere are not simply due to competition for scarce energy, but also due to the fact that only a few lineages (e.g., parids, corvids, woodpeckers, finches) have evolved adaptations to seek out scarce winter resources. In other words, some consideration of evolution is needed in this paper, which at present adopts a purely ecological view.

Figure S1 is also relevant to this point. It shows that the current richness of birds during the non-breeding season is underestimated. That is, the model predicts there are no birds found in the colder latitudes of the northern hemisphere in their simulated world. I suspect this is because the model does not capture the fact that these birds have specialized adaptations that sets them apart ecologically and physiologically from the rest (see my point above) and allows them to survive in energy depleted environments. Thus, I think the paper could do a better job of mentioning that by treating all species as ecologically equivalent they are missing nuances of evolutionary history that influence species' ecological interactions with each other and with energy patterns. They do mention some deficiencies of the model in capturing ecology (e.g., they discuss the model's low species richness in tropical mountains), but again, evolution is scarcely considered as a factor.

Finally, I think it is worth discussing more explicitly the extent to which the model captures a tradeoff between the difficulty of surviving a harsh winter with low available energy, and the energy required for migrating. I suspect this is included implicitly in the model, but I had some trouble identifying it. For example, in lines 308-348, and in the caption of Figure 1, I could use more explanation of how cold winter temperatures and low NPP tradeoff with the modelled cost of migration when assigning species to hexagons.

Detailed/minor comments:

Line 23 and Line 116: Reference #17 seems to be out of place here, I think it is a mistake? Line 116 also misses some important references, including:

Louchart, A. (2008). Emergence of long distance bird migrations: a new model integrating global climate changes. *Naturwissenschaften* 95, 1109–1119.

Bruderer, B. & Salewski, V. (2008). Evolution of bird migration in a biogeographical context. *Journal of Biogeography* 35, 1951–1959.

Line 47: “...recent analyses of global bird migration have suggested that migration is driven not just by species specific climatic niches but by inter-specific competitive interactions^{22,23}” This is an example of over-interpretation the previous model (ref 22). Given that these models treat all species as ecologically equivalent and do not consider traits, I don’t think that one of the results of this or the previous paper is that migration is driven by “competitive interactions”. Of course, migration is likely driven by competitive interactions to some degree, I just don’t think that follows very well from the model results.

Line 49 and 52: The term “mechanistic model” is introduced here, but not explained until much later in the paper. It needs more explanation here, and I think it should also clarify that the study is a simulation.

Line 84: “inter-specific competition for maximising energetic fitness appears to be at the core of bird migration”...This statement seems rather casual/flippant to me. At the core of what aspect of bird migration? More nuanced language is needed to understand the authors’ point here.

Line 112-114: “Our findings suggest that throughout the last 50,000 years, spanning the last glacial maximum (~20,000 years ago), bird migration remained an important global phenomenon, contradicting the hypothesis that migration is mainly a phenomenon of inter-glacial periods.” This important and interesting result directly refutes a recent paper by Zink and Gardner (2017) in *Science Advances* that attempted to push the hypothesis that migration is mainly a phenomenon of inter-glacial periods. That paper, as well as several earlier ones suggesting the same idea, should be mentioned, given the contradiction presented by this new analysis.

Line 136-139: This definition of a “mechanistic model” needs to go earlier in the paper (see comment above). Feels out of place here.

Line 171-174: Interesting idea about flight calls, but feels a bit tossed in here at the end. I’d rather see the space used to discuss more nuances of their results.

Line 263: Where did this equation come from? And why $u = 65$? Some modeling choices seemed a bit arbitrary, which is probably fine, but needs more explanation.

Reviewer #2 (Remarks to the Author):

This paper explores further a model of bird distributions (recently published in Nature E&E) with respect to the propensity to migrate, and relate these propensities to models of climate change over the last 50K years. The paper also reports comparisons between the two main migration systems in the world, i.e. the New World and the Palearctic-African systems. The basics of the model are a number of assumptions about energy minimization of migratory movements and net energy investment in reproduction.

The authors chose to minimize energy cost of moving the shortest distance between breeding and wintering sites. This is one of alternative currencies, while it has become increasingly evident that many birds migrate to minimize the time of migration (this strategy is more wasteful with energy) and may/may not lead to different conclusions.

The main finding is that the number of migrants has varied in relation to the climate change glaciation history, with some differences between the two migration systems. The results are not surprising as such, but the problem is that the predictions derived from the simulations are hardly testable. The authors concede that behaviours like migration are not represented in the fossil record, and for this reason we will be left with a number of untestable predictions. There are some agreement with current distributions of birds, but the correlations (and hence r-square values) are rather low, with proportion of variation explained by the model simulations falling in the range 36-60%. Hence, there are lots of variation not explained. The conclusion that these data could be used to understand the past "functional roles of birds in communities and ecosystems" seems somewhat optimistic.

At the end the authors identify a number of improvements to the model that would make it more realistic. One wonders why these were not implemented before writing up this paper, which now looks more like a progress report of an ongoing study.

Reviewer #3 (Remarks to the Author):

As a climate expert, my review only focused on the parts of the manuscript that referred to climate, and will accordingly be short. I am unable to properly evaluate the findings of this work.

Climate simulations of the past 50k years are extremely rare, and as such the authors have made a reasonable choice by opting for the HAD-CM3 model. It is probably one of the best long-term simulations available. The down-scaling approach used is also sound.

The ice-sheet model used in this paper is outdated. The ICE-6G model by Peltier is publicly available. However, I do not think it would make any significant difference with the current results as most of the changes between the 5G and 6G versions are changes in elevation and distribution of ice, not about spatial extension. I would still recommend updating.

An interesting point of comparison for this study would be to also look at the TRACE-21K global simulation. While much shorter than the one used in this study, it still covers the period from the LGM to present day. The climate simulations are certainly different at a local scale, but it would be interesting to see if the same global patterns of bird migration can be extracted. It would provide strong support to the present results.

Reviewer #1

This is an interesting and well-written paper that tests whether or not birds were likely to have been migratory throughout the last 50,000 years of earth history, given shifting climatic conditions and glacial maxima. The research builds on a previously published model used to simulate the seasonal geographic distributions of birds in relation to global energy supply, and uses present-day model predictions to infer past conditions. The extent to which birds migrated during the last glacial maximum has been the subject of some recent debate, and this paper is by far the most sophisticated analysis of the question to date.

We are pleased that the reviewer sees our manuscript as interesting and valuable.

General comments:

1) I liked the paper and generally found it readable and accessible, with a few exceptions that I note below. My main comment is that the entire modeling framework is quite complex as a whole, and relies on a large number of moving parts and building blocks. Some of these parts are well-described in this paper, and for others the reader has to look back to the previous paper describing the original model (which I did). Some of the methods used are intuitive, and others are very specialized methods with which a typical reader is unlikely to be familiar (e.g., ‘spreading-dye algorithm,’ ‘earth’s-mover’s distance,’ the ‘delta-method.’). I applaud the author’s creativity and ingenuity to find methods that they could apply to each component of their model, and also their efforts to look to the biological literature to come up with best-guess parameter estimates. They have made a strong effort to build a model to test their hypotheses in a reasonable and sophisticated way. Nevertheless, putting everything together amounts to a black box in which it is essentially impossible for a reader to understand how particular modeling or methodological decisions affects their results. Their sensitivity analyses help with this to some extent, but not entirely. This complexity is not necessarily a bad thing, but it demands some care when describing and interpreting the study.

We agree that the model relies on several building blocks and uses various specialized methods, which carries the risk of being seen as a black box by the reader. To make it clearer to the reader, and also to address other comments by the reviewer, we modified how we describe the model in the main text by adding substantially more information when we describe it in the introduction and results sections (lines 54–87). We also give more information in the caption of Fig 1, which describes in details how the model works (i.e. component by component) in addition to the methods section that provides much details on the new improvements in relation to the previous version of the model (i.e. section on maximising energetic fitness). However, it is not our aim here to describe the model with the same level of details as in the original paper, for both space-constraints and redundancy reasons. We hope to have now made it clearer to the reader that despite involving some complexity in its design, this remains a simple mechanistic model – based on simple rules with a limited number of parameters and only one free parameter that could not be estimated directly from the literature. Furthermore, as the reviewer mentions, we conducted a

sensitivity analysis of the model, in which we varied several parameters and building blocks in order to explore how the model behaviour responds. Through these, we believe that we did our best efforts to deal with, and communicate, the relative complexity of our approach while avoiding redundancy with the original paper describing the model.

What I think is mainly needed now is more contextualization of their study as a simulation, because essentially that's what they're doing -- simulating a virtual world. So, for example, Line 8 in the abstract should read, "we apply a mechanistic modelling framework to climatic reconstruction data to SIMULATE the effect of major climatic changes on the spatial patterns..." To me, this gives a better description to the reader of what is really happening. Similarly, throughout the paper the authors should be more careful not to take the results of the original model (reference 22) as established fact, because, again, that study is a simulation and in fact the current model is pitched as an improvement over the previous one. I should be clear that what I'm suggesting are relatively minor changes to descriptive language, but I think they are important.

We think this is a very good point that will improve the readability of the manuscript. We follow the reviewer's suggestion and now emphasise earlier in the manuscript that this is a simulation study, including: in the abstract (line 6); when we describe the mechanistic model (lines 54–87, 103, 107, 301); and throughout the text when discussing the model results (lines 98, 145, 147, 164). We also edited the text to remove any ambiguity that the results of the model (both the original version and the new version presented here) could be seen as established facts rather than simulation results (e.g. lines 48, 144–146, 164–166).

2) I think the authors should consider that migration is just one of several adaptations that allow for organismal persistence in a seasonal environment and think about how this influences their interpretation (see, for example, Winger et al. 2019 Biological Reviews (doi: 10.1111/brv.12476), for a detailed discussion of this). The reason this is important to discuss is that alternative adaptations to seasonality (e.g., hibernation of mammals, deciduous trees dropping leaves) lead to a different distribution of biodiversity throughout the year, and yet all of these organisms are dealing with the same seasonal flux and spatial distribution of global energy.

We agree, and in fact this same point was made in the paper describing the original model when discussing how the model could be generalized to non-mobile taxa with other adaptations to seasonality (Somveille et al. 2018 Nat Ecol Evol – penultimate paragraph in the discussion). Following the reviewer's suggestion (also see below) we have edited the text in the present manuscript to make the point that birds too can have other adaptations to seasonality besides migration (lines 164–166, 171–180).

Below, I point out a couple of specific parts of the text that would be improved by considering this interpretation of migration as one type of persistence strategy:

Line 133: "In our reconstruction of global bird distributions, migration appears as an

emergent phenomenon resulting from the maximisation of species' energetic fitness, with spatial diversity patterns reflecting an equilibrium between the distribution of the global avifauna and climate." I think this characterization needs further thought. The results, in my opinion, do not show migration itself as an emergent phenomenon of energetic fitness per se. Rather, the seasonal distributional patterns of birds are the emergent response to energy, given that birds use migration as their principal strategy for maximizing energetic fitness. I.e., if more lineages of birds had evolved alternative strategies for coping with low energy environments, their distributional patterns would be different. The depauperate winter avifaunas of the northern hemisphere are not simply due to competition for scarce energy, but also due to the fact that only a few lineages (e.g., parids, corvids, woodpeckers, finches) have evolved adaptations to seek out scarce winter resources. In other words, some consideration of evolution is needed in this paper, which at present adopts a purely ecological view.

Our point is that migration is not something that is added to the model as an input but it is instead something that emerges from the simple rules we have used to build the model. It is however true that the possibility of migration is something that is intrinsic to the model, with the end result being that many of the virtual species end up as migrants, their predicted patterns matching well the empirical patterns. The model does not allow for other strategies for coping with seasonality, and accordingly none emerge. We have now rephrased this sentence to read: "In our simulations, patterns in the seasonal redistribution of the world's avifauna emerge from the optimisation of energy budgets as birds use migration as a strategy to maximise energetic fitness" (line 164–166). We have also followed the reviewer's advice by including a consideration of evolutionary processes allowing persistence despite scarce winter resources (lines 171–180).

Figure S1 is also relevant to this point. It shows that the current richness of birds during the non-breeding season is underestimated. That is, the model predicts there are no birds found in the colder latitudes of the northern hemisphere in their simulated world. I suspect this is because the model does not capture the fact that these birds have specialized adaptations that sets them apart ecologically and physiologically from the rest (see my point above) and allows them to survive in energy depleted environments. Thus, I think the paper could do a better job of mentioning that by treating all species as ecologically equivalent they are missing nuances of evolutionary history that influence species' ecological interactions with each other and with energy patterns. They do mention some deficiencies of the model in capturing ecology (e.g., they discuss the model's low species richness in tropical mountains), but again, evolution is scarcely considered as a factor.

This is a very good point. Following the reviewer's suggestion, we have now explicitly made this point in the manuscript (lines 171–180).

Finally, I think it is worth discussing more explicitly the extent to which the model captures a tradeoff between the difficulty of surviving a harsh winter with low available energy, and the energy required for migrating. I suspect this is included implicitly in the model, but I had some trouble identifying it. For example, in lines 308-348, and in the caption of Figure 1, I

could use more explanation of how cold winter temperatures and low NPP tradeoff with the modelled cost of migration when assigning species to hexagons.

This interplay between energy assimilation (access to resources at each season) and energy used for travelling is the core mechanism of our model. The model essentially balances these two processes to assign species to optimal seasonal geographical distributions. To make it clearer to the reader, and as suggested by the reviewer, we added a sentence in the caption of Figure 1 (lines 639–642) and in the text (lines 366–368).

Detailed/minor comments:

Line 23 and Line 116: Reference #17 seems to be out of place here, I think it is a mistake?

Yes it is a mistake. Thank you for spotting this. This reference was intended for line 21. We corrected it.

Line 116 also misses some important references, including:

Louchart, A. (2008). Emergence of long distance bird migrations: a new model integrating global climate changes. *Naturwissenschaften* 95, 1109–1119.

Bruderer, B. & Salewski, V. (2008). Evolution of bird migration in a biogeographical context. *Journal of Biogeography* 35, 1951–1959.

We included more references here, including the two suggested by the reviewer.

Line 47: “...recent analyses of global bird migration have suggested that migration is driven not just by species specific climatic niches but by inter-specific competitive interactions^{22,23}” This is an example of over-interpretation the previous model (ref 22).

Given that these models treat all species as ecologically equivalent and do not consider traits, I don’t think that one of the results of this or the previous paper is that migration is driven by “competitive interactions”. Of course, migration is likely driven by competitive interactions to some degree, I just don’t think that follows very well from the model results.

Here, by “competitive interactions” we mean inter-specific competition for access to resources (i.e. in an environment with a carrying capacity) that does not necessarily require differences between species (e.g. interference competition). This process was explicitly included and tested in the original model (and in the version presented here too) and led to a high explanatory power, hence “suggesting” that it is an important mechanism.

To clarify this in the text, we changed the sentence to: “...recent simulation analyses of global bird migration^{23,24} have suggested that, in addition to species-specific climatic niches, the seasonal redistribution of species is shaped by inter-specific competition for access to limited resources, for example associated with mutual interference^{31,32}, increasing search time³³ and territorial defence³⁴” (lines 48–52).

Line 49 and 52: The term “mechanistic model” is introduced here, but not explained until

much later in the paper. It needs more explanation here, and I think it should also clarify that the study is a simulation.

We amended the text and included a more substantial description of the mechanistic model at this point of the manuscript, including a clarification that the study is a simulation (lines 54–87).

Line 84: “inter-specific competition for maximising energetic fitness appears to be at the core of bird migration”... This statement seems rather casual/flippant to me. At the core of what aspect of bird migration? More nuanced language is needed to understand the authors’ point here.

Inter-specific competition for access to limited resources (which does not require differences between species) is explicitly included in our model and lead to a high explanatory power, which suggests that it is an important mechanism (see our response two comments above). We nonetheless agree that a somewhat more nuanced language is needed and so we changed the sentence to: “The model’s good performance at simulating current breeding and non-breeding patterns of the global migratory avifauna supports an important role of energy efficiency (i.e. optimising the interplay between energy assimilation, which is affected by inter-specific competition for access to resources, and the energetic cost of travelling) in driving bird migration.” (lines 98–102).

Line 112-114: “Our findings suggest that throughout the last 50,000 years, spanning the last glacial maximum (~20,000 years ago), bird migration remained an important global phenomenon, contradicting the hypothesis that migration is mainly a phenomenon of inter-glacial periods.” This important and interesting result directly refutes a recent paper by Zink and Gardner (2017) in Science Advances that attempted to push the hypothesis that migration is mainly a phenomenon of inter-glacial periods. That paper, as well as several earlier ones suggesting the same idea, should be mentioned, given the contradiction presented by this new analysis.

We thank the reviewer for pointing us to this study. Our results indeed contradict directly what they found for 56 North American species using species-specific climatic niche models. We added this reference line 49 and added a couple of sentences to contrast our results and approach to theirs (lines 138–143).

Line 136-139: This definition of a “mechanistic model” needs to go earlier in the paper (see comment above). Feels out of place here.

Agreed. We have now expanded our presentation of the mechanistic model earlier in the text (lines 54–87) and we have moved this definition to the Model Overview section in the Methods (lines 251–254).

Line 171-174: Interesting idea about flight calls, but feels a bit tossed in here at the end. I’d

rather see the space used to discuss more nuances of their results.

We kept this sentence advancing this idea about flight calls as we had enough space to expand and nuance our discussion of the results in the previous paragraphs.

Line 263: Where did this equation come from? And why $u = 65$? Some modeling choices seemed a bit arbitrary, which is probably fine, but needs more explanation.

This equation is used to transform NPP values (i.e. in each hexagon) into a carrying capacity (i.e. number of species that can co-exist in each hexagon – note that all species have the same local abundance in our model). To do that, NPP values were log-transformed and multiplied by the constant μ , which was set to 65 in order to obtain realistic simulated magnitudes of species richness. This was arbitrarily chosen but the sensitivity analysis (which varies μ) indicates that it does not affect the performance of the model in recreating the patterns in the geographical variation of species richness (Fig S4). We have now added this explanation in the methods section (lines 300–303).

Reviewer #2

This paper explores further a model of bird distributions (recently published in Nature E&E) with respect to the propensity to migrate, and relate these propensities to models of climate change over the last 50K years. The paper also reports comparisons between the two main migration systems in the world, i.e. the New World and the Palearctic-African systems. The basics of the model are a number of assumptions about energy minimization of migratory movements and net energy investment in reproduction.

The authors chose to minimize energy cost of moving the shortest distance between breeding and wintering sites. This is one of alternative currencies, while it has become increasingly evident that many birds migrate to minimize the time of migration (this strategy is more wasteful with energy) and may/may not lead to different conclusions.

We agree that our approach to modelling migration costs is highly simplistic, and that in reality the most time-efficient – but also energy-efficient – migration strategies often might differ from the shortest distance. We included a new paragraph discussing these limitations and the potential other mechanisms to be included (lines 164–184). Nonetheless, we believe that the shortest distance is a reasonable proxy, given the scale at which our analysis is conducted, our focus on seasonal species distributions rather than migration trajectories and that we aim to use a mechanistic model based on simple rules. If anything, using such a simplistic approach should have reduced the predictive capacity of our model, so the fact that it nonetheless produced simulated patterns very similar to empirical patterns indicates that the proxy we have used for modelling migration costs is generally robust.

The main finding is that the number of migrants has varied in relation to the climate change glaciation history, with some differences between the two migration systems. The results are

not surprising as such, but the problem is that the predictions derived from the simulations are hardly testable. The authors concede that behaviours like migration are not represented in the fossil record, and for this reason we will be left with a number of untestable predictions. There are some agreement with current distributions of birds, but the correlations (and hence r-square values) are rather low, with proportion of variation explained by the model simulations falling in the range 36-60%. Hence, there are lots of variation not explained. The conclusion that these data could be used to understand the past “functional roles of birds in communities and ecosystems” seems somewhat optimistic.

Our study uses a more sophisticated and integrative approach than previous work and our results contradict the hypothesis (notably advanced by Zink & Gardner 2017) that most migratory species were sedentary during the last glacial maximum and that glaciations are major “migratory switches” (lines 138–146), while we also found regional variations in how bird migration responded to past climate change. As we note in the text, we do not have empirical data on the migration patterns of birds in the past for validating our simulated reconstructions of global bird migration. However, we believe that attempting to explore this question using a novel approach that is particularly appropriate for making predictions under different environmental conditions such as the LGM (i.e. mechanistic model at the avifauna scale) is worthwhile as it generates hypotheses and provides a “best estimate” of how global bird migration has evolved over the past 50,000 years. We argue that the correlations between the simulated and empirical patterns are actually remarkably high (0.618–0.801; Fig S1) for such a simple model (only one free parameter) attempting to recreate how all the migratory land birds distribute seasonally across the world using simple rules reflecting key ecological mechanisms. Moreover, the model predicts a stable number of simulated species over the past 50,000 years (lines 131–135), which is realistic. We therefore believe that the confidence in our model’s performance is high enough for making valuable reconstructions of global bird migration. Having said that, we agree with the reviewer that a significant amount of the variation is not explained by our model and that this must be considered in our discussion and interpretation of the results. We added a paragraph discussing the limitations of our model and potentially missing mechanisms (lines 164–184). The sentence “Understanding how the importance, prevalence and magnitude of migration across the avifauna vary throughout glacial cycles has relevance not just for understanding migration as a behavioural phenomenon, but also for gauging the past seasonal dynamics and the functional roles of birds in communities and ecosystems” is not a conclusion of our study but rather a motivation for doing it. To make it clearer to the reader, we moved this sentence earlier in the text (lines 40–43).

At the end the authors identify a number of improvements to the model that would make it more realistic. One wonders why these were not implemented before writing up this paper, which now looks more like a progress report of an ongoing study.

We originally identified one main avenue to build upon this work: combining the model presented in this study (which is at avifauna-scale) with species-level distribution models in order to investigate how species-specific seasonal ranges have evolved since the LGM and

explore which species might have been resident for example (lines 185–194). Such development would be a major study on its own and would allow tackling a different set of research questions on the origin and evolution of migration. Our manuscript is therefore a standalone study addressing how the global migratory avifauna has changed over the past 50,000 years. In revising the manuscript, we now include a paragraph explicitly discussing the limitations of our model and the mechanisms that could be potentially added (lines 164–184). This discussion, however, does not suggest that this paper is a “report” of a larger ongoing study since (i) these additions to the model are substantial studies on their own (e.g. modelling individual-level migratory trajectories explicitly; including other currencies to optimise such as time and predation risk; or including other evolutionary adaptations than migration to deal with seasonality); and (ii) we here intentionally build a simple model based on energy efficiency at the avifauna level so we do not intend to make it over-complex with including many mechanisms and scales simultaneously.

Reviewer #3

As a climate expert, my review only focused on the parts of the manuscript that referred to climate, and will accordingly be short. I am unable to properly evaluate the findings of this work.

Climate simulations of the past 50k years are extremely rare, and as such the authors have made a reasonable choice by opting for the HAD-CM3 model. It is probably one of the best long-term simulations available. The down-scaling approach used is also sound.

The ice-sheet model used in this paper is outdated. The ICE-6G model by Peltier is publicly available. However, I do not think it would make any significant difference with the current results as most of the changes between the 5G and 6G versions are changes in elevation and distribution of ice, not about spatial extension. I would still recommend updating.

We thank the reviewer for pointing this out. We updated the ice-sheet model to ICE-6G and re-ran the analysis. As the reviewer suspected, no significant differences were obtained with the updated ice-sheet model compared to the previous results. We updated all the figures as well as the methods section and the analysis values presented in the main text.

An interesting point of comparison for this study would be to also look at the TRACE-21K global simulation. While much shorter than the one used in this study, it still covers the period from the LGM to present day. The climate simulations are certainly different at a local scale, but it would be interesting to see if the same global patterns of bird migration can be extracted. It would provide strong support to the present results.

We agree that using also TRACE-21K would provide an interesting point of comparison and would be a good way to investigate whether the model results are robust to different climate data. However, this model only outputs annual NPP values, rather than the monthly values that are required for analysing bird migration (we contacted the National Center for

Atmospheric Research, which hosts the outputs of TRACE-21K, and they confirmed that “only annual NPP data are available from TraCE”). As TRACE-21K uses an advanced process-based vegetation model (LPJ Guess), it might be possible to modify the source code to allow this model to generate monthly values, but this is beyond the scope and timeframe of this study and would require a substantial extension of time for allowing us to do that.